



# Sensitivity of Future Projections of the Wilkes Subglacial Basin Ice Sheet to Grounding Line Melt Parameterizations

Yu Wang[1], Chen Zhao[1], Rupert Gladstone[2], Thomas Zwinger[3], Benjamin K. Galton-Fenzi[4, 1, 5], and Poul Christoffersen[5, 1]

[1]Australian Antarctic Program Partnership, Institute for Marine and Antarctic Studies, University of Tasmania, Hobart, Australia
[2]Arctic Centre, University of Lapland, Rovaniemi, Finland
[3]CSC-IT Center for Science, Espoo, Finland
[4]Australian Antarctic Division, Kingston, Australia
[5]Australian Centre for Excellence in Antarctic Science, University of Tasmania, Hobart, Australia

**Correspondence:** Yu Wang (yu.wang0@utas.edu.au)

**Abstract.** Projections of Antarctic Ice Sheet mass loss and therefore global sea level rise are hugely uncertain, partly due to how mass loss of the ice sheet occurs at the grounding line. The Wilkes Subglacial Basin (WSB), a vast region of the East Antarctic ice sheet, is thought to be particularly vulnerable to deglaciation under future climate warming scenarios. However, future projections of ice loss, driven by grounding line migration, are known to be sensitive to the parameterisation of ocean-induced basal melt of the floating ice shelves, and specifically, adjacent to the grounding line - termed Grounding Line Melt Parameterizations (GLMPs). This study investigates future ice sheet dynamics in the WSB with respect to four GLMPs under both the upper and lower bounds of climate warming scenarios from the present to 2500, with different model resolutions and choices of sliding relationships. The variation in these GLMPs determines the distribution and the amount of melt applied in the finite element assembly procedure on partially grounded elements (i.e., elements containing the grounding line). Our findings indicate that the GLMPs significantly affect both the trigger-timings of tipping points and the overall magnitude of ice mass loss. We conclude that applying full melting to the partially grounded elements, which causes melting on the grounded side of the grounding line, should be avoided under all circumstances due to its poor numerical convergence and substantial overestimation of ice mass loss. We recommend preferring options that depend on the specific model context, either 1) not applying any melt immediately adjacent to the grounding line or 2) employing a sub-element parameterisation. Based on our best model results, a tipping point is projected to occur between 2200 and 2300, leading to massive and rapid retreat across the WSB and a significant increase in ice discharge from 200 to 500 $\mathrm{Gt\,a^{-1}}$. In this context, our simulations suggest that the WSB ice sheet could contribute between 0.23 to 0.34 m to global sea level rise by 2500.

## 1 Introduction

Melting beneath ice shelves and iceberg calving are recognised as equally important contributors to the current mass loss of the Antarctic Ice Sheet (Greene et al., 2022) , accounting for a total contribution of approximately 5.2 mm to global sea level rise since 2003 (Smith et al., 2020). Basal melting plays a crucial role in the contemporary amplification of ice discharge in



Antarctica (Noble et al., 2020; Adusumilli et al., 2020). Variations in basal melt rates exert significant influence on ice shelf thickness, with thinning leading to a diminished ice shelf buttressing effect. The reduction in buttressing subsequently results in the acceleration of ice streams that supply the ice shelf. Such acceleration contributes to dynamic thinning of the ice upstream

of the grounding line, inducing grounding line retreat. The associated loss of basal resistance may, in turn, provoke a positive feedback if the subglacial topography deepens towards the interior of the continent. This unstable behaviour is known as the Marine Ice Sheet Instability (MISI) (Schoof, 2007; Favier et al., 2014; Robel et al., 2019).

The Wilkes Subglacial Basin (WSB; Fig. 1), located west of the Trans-Antarctic Mountains in East Antarctica, spans approximately 400,000 $km^2$, with depths extending as far as 2000 m below sea level in a deep marine-based setting. Ice flow

predominantly occurs along two deep troughs extending subglacially towards the Cook and Ninnis Ice Shelves, which currently discharge $40.6\,\mathrm{Gt\,a^{-1}}$ and $23.0\,\mathrm{Gt\,a^{-1}}$ of ice into the ocean, respectively (Rignot et al., 2019, Fig. 1). The WSB is notable for its extensive ice reserves and vulnerability to Marine Ice Sheet Instability (Crotti et al., 2022; Mengel and Levermann, 2014). A tipping point behaviour (onset and continuation of MISI) has been shown to occur in simulations (Sutter et al., 2020; Mengel and Levermann, 2014), yet there is a paucity of observations and modelling efforts to inform this potentially unstable behaviour

(Golledge et al., 2017). As such, the WSB may be particularly sensitive to melting beneath the ice shelf and the grounding line dynamics, thereby rendering the disparities among our sensitivity experiments more pronounced. These factors motivated us to select the WSB as the focus of our study.

Recent studies indicate that the migration of the grounding line is extremely sensitive to how basal melt occurs adjacent to the grounding line (Parizek et al., 2013; Arthern and Williams, 2017; Reese et al., 2018; Goldberg et al., 2019). However, due

to constrained observations, our understanding of the actual melt rates at the grounding line and their underlying mechanisms remains in its infancy (Robel et al., 2022). Traditional plume and ocean models generally predict that the basal melt rates tend to approach zero near the grounding line (e.g. Galton-Fenzi, 2009; Lazeroms et al., 2018; Cornford et al., 2020; Burgard et al., 2022), with the peak melt occurring about 10 to 15 km away from it (Slater et al., 2017, 2020). In a detailed study, Burgard et al. (2022) applied the ocean model, NEMO, to simulate Antarctic ice shelf melt rates, finding more than half of

the ice shelves show melt rates approximating zero at the grounding line, with an average rate of $0.45\,\mathrm{m\,a^{-1}}$ across all of them. Nevertheless, other studies challenge this traditional understanding represented by the plume model. Robel et al. (2022) discussed the possibility of high melting at, and even glaciologically upstream of, the grounding line caused by the intrusion of layered warm salty water. In their theoretical model experiments, seawater intrudes as far as several kilometers upstream of the grounding line, potentially doubling ice mass loss (Robel et al., 2022). Ciracì et al. (2023) validated the seawater intrusion

theory by analysing satellite radar interferometry, revealing up to $80\,\mathrm{m\,a^{-1}}$ melt rates in the tidally influenced grounding zone of Petermann Glacier in Greenland. From another perspective, the Antarctic basal melt rates computed by Adusumilli et al. (2020), based on remote sensing observations and ice flux divergence calculation, do not show a pattern of melt rates approaching zero at the grounding line.

Modelling studies suggest that ice sheet models are more sensitive to melt rates near the grounding line than to cavity-

integrated melt rates beneath ice shelves (e.g. Gagliardini et al., 2010; Reese et al., 2018; Morlighem et al., 2021). As such, accurately simulating melt patterns, particularly near the grounding line, might be at least as important as simulating a realistic



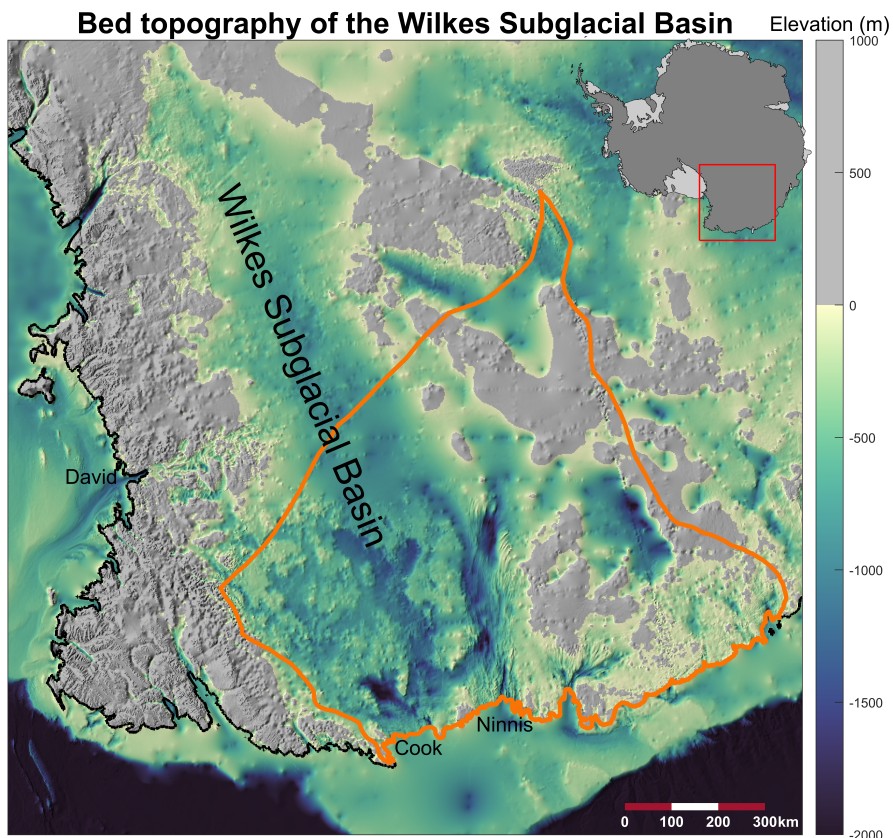

**Figure 1.** Bed topography of the Wilkes Subglacial Basin (WSB) and the designated catchment used as the model domain. The three primary outlet glaciers of the WSB, (Cook, Ninnis and David glaciers) are marked. The orange contour delineates the model domain in this study.

integrated melt (Burgard et al., 2022). Accurate representation of basal melt at the grounding line is crucial for ice flow models to reduce uncertainties in forecasting ice sheet dynamics and future mass loss (Seroussi and Morlighem, 2018). However, due to the discretisation of the ice sheet model, there inevitably exist grid cells or elements at the grounding line where ice is partially grounded and partially floating. How to represent basal melting within these cells remains a challenging and unresolved issue, which is further explored here.

In the past decade, various parameterisation schemes for handling sub-grid scale features at the grounding line in basal friction and melt have been explored (e.g. Gladstone et al., 2010; Leguy et al., 2014; Seroussi et al., 2014; Feldmann et al., 2014; Arthern and Williams, 2017; Leguy et al., 2021). The initial motivation to explore grounding line parameterization was to optimise the treatment of basal friction at the grounding line, given its high impact on grounding line dynamics (Seroussi et al., 2014). Sub-element parameterizations for the representation of basal friction generally over partially grounded elements provide improved convergence of model behaviour with finer mesh resolution (Leguy et al., 2014; Seroussi et al., 2014; Feldmann et al., 2014), and they are widely used in subsequent research on ice sheet modelling (e.g. Seroussi et al., 2019, 2020;



Nowicki et al., 2020). Seroussi and Morlighem (2018) pioneered a comprehensive study on representation of basal melt under
partially floating cells, based on the MISMIP model configuration (Asay-Davis et al., 2016). They recommend models should
avoid applying melt rates over entire partially floating cells, as this gives worse convergence with resolution and overestimates
grounding line retreat at typically used resolutions (Seroussi and Morlighem, 2018). Following this, a majority of subsequent
ice sheet modelling efforts adopt melt parameterizations assuming zero melt at the grounding line (Seroussi et al., 2019, 2020).
In ice sheet model intercomparisons, initMIP-Antarctica (Seroussi et al., 2019), it was found that marine ice sheet models using
sub-element melt (SEM) parameterizations are consistently more sensitive to ocean forcing than those without melt applied
to these elements (increasing the Antarctic contribution to sea level rise by 50 %–100 %; Seroussi et al., 2019). However,
recent studies (Leguy et al., 2021; Berends et al., 2023) suggests that, in their finite-difference based model experiments,
models applying melt at the grounding line on the partially floating cells overall outperform those not applying melt in terms
of convergence with resolution.

This study seeks to delve deeper into various parameterization solutions for basal melt at the grounding line applied to
the domain of the Wilkes Subglacial Basin through a series of sensitivity experiments. Additionally, we provide quantitative
projections of future ice mass loss based on the most credible model configurations identified through our analysis. We detail
the methods in Sect. 2, including the model configuration, historical run and experimental design. Results are presented in Sect.
3, with a subsequent discussion in Sect. 4. Conclusions are provided in Sect. 5.

## 2 Methods

We use Elmer/Ice (Gagliardini et al., 2013) to conduct a series of ice sheet simulations for the WSB. Elmer/Ice is an open-source
finite-element, ice sheet/shelf model, capable of solving the full-Stokes equations but also allows for various simplifications,
such as the Shallow Shelf Approximation we use here (SSA; MacAyeal, 1989). We conduct a series of sensitivity experiments
of the WSB with SSA to investigate the sensitivity of grounding line movement and ice mass loss to different Grounding Line
Melt Parameterizations (GLMPs) under future climate forcing scenarios. The workflow is illustrated in Fig. 2. The following
subsections will detail the model setup and inversions (Sect. 2.1), and transient simulations (Sect. 2.2), including historical and
future runs, sequentially.

### 2.1 Model setup and inversions

The two-dimensional (2-D) mesh used for the WSB domain is constructed using Gmsh (Geuzaine and Remacle, 2009). It
features a quasi-uniform, unstructured triangular grid at a 1 km resolution. The inland domain boundary defining the glacier
basin of the WSB model is sourced from MEaSUREs Antarctic Boundaries, Version 2 (Mouginot et al., 2017; Rignot et al.,
2013). The coastline boundary is derived from MEaSUREs BedMachine Antarctica, Version 3 (Morlighem, 2022; Morlighem
et al., 2020). The locations of calving front and inland boundary are held fixed throughout the simulations. We then conduct
mesh refinement using Mmg (Dapogny et al., 2014) to optimise computational efficiency without compromising accuracy. We
estimate the location of the grounding line in the year 2300, based on the projected grounding line movement under the most



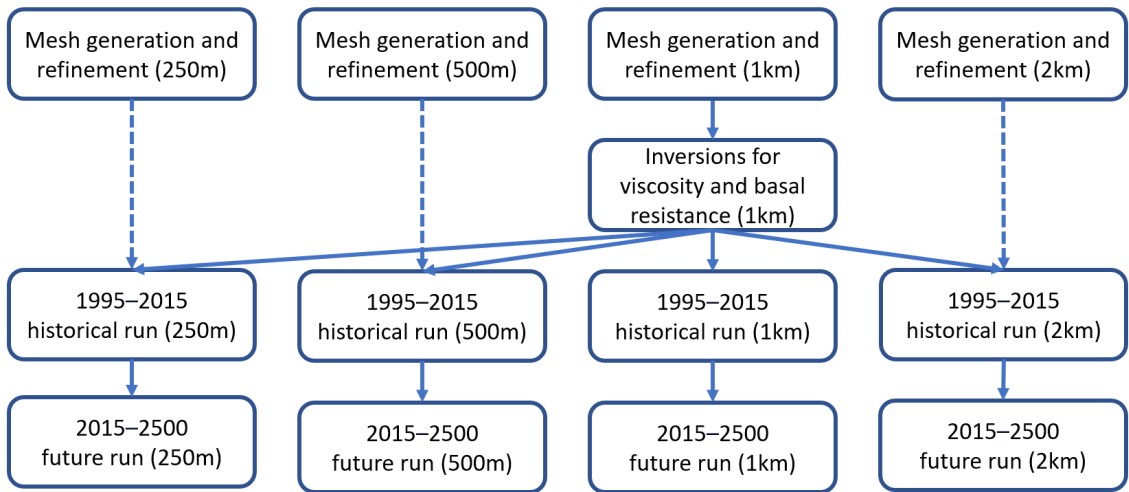

**Figure 2.** Overview of the experiments workflow in this study. Marked in parentheses is the resolution of the model grid. The results obtained from the inversion, including basal friction parameter $\beta$ and viscosity enhancement factor $E_\eta$ are interpolated onto four grids respectively to initialise the subsequent historical runs.

severe ice loss scenario from the Antarctic model in the ISMIP6–2300 project (Seroussi, 2022). For the area downstream of this line, the grid is refined to characteristic resolutions of 250 m, 500 m, 1 km, and 2 km, respectively (Fig.3), in preparation for subsequent sensitivity experiments. Conversely, for its upstream inland region, the mesh resolution is progressively transitioned to coarser scales. The four grids maintain a very similar mesh resolution in the far inland area, characterised by elements of

approximately 17 km horizontal extent. This refinement strategy is designed to prevent the grounding line from retreating into areas of coarser resolution during centennial-scale transient runs. Besides, the local refinement metric draws upon both ice surface velocity observations (Mouginot et al., 2019a, b) and ice thickness (Morlighem, 2022; Morlighem et al., 2020), allocating slightly finer resolution in regions with pronounced gradients in velocity and thickness. The statistics of the four grids are shown in Table 1.

**Table 1.** Summary of the four grids.

| Mesh resolution | Nodes | Triangular elements |
|---|---|---|
| 2 km | 54 771 | 94 894 |
| 1 km | 172 389 | 316 170 |
| 500 m | 612 204 | 1 142 726 |
| 250 m | 2 317 821 | 4 270 368 |

In this study, we solve the 2-D vertically integrated SSA equations (MacAyeal, 1989) for the stress balance. We consider two friction laws for the basal shear stress, $\boldsymbol{\tau}_b$, the linear Weertman law (Weertman, 1957) and regularised Coulomb law (Joughin




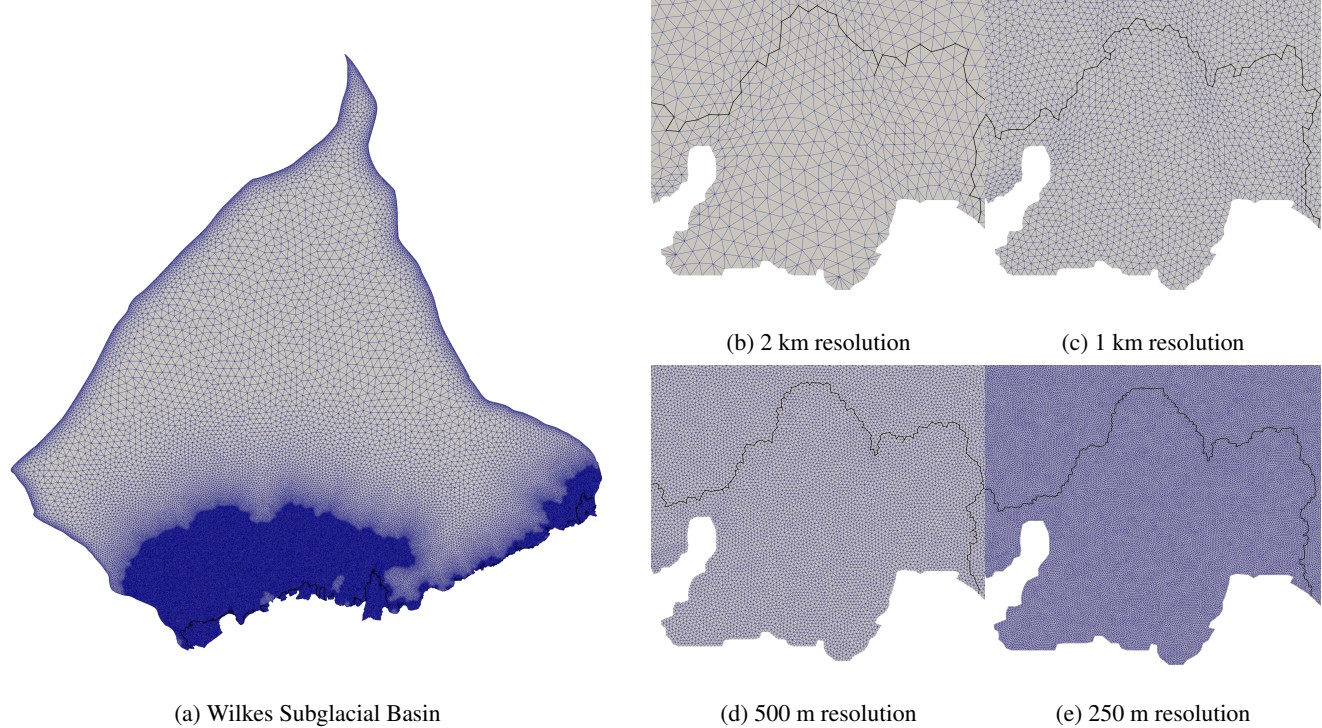

(a) Wilkes Subglacial Basin

(b) 2 km resolution

(c) 1 km resolution

(d) 500 m resolution

(e) 250 m resolution

**Figure 3.** Refined grid for the Wilkes Subglacial Basin with 1 km characteristic mesh resolution(a); Grid details at the Ninnis Ice Shelf with 2 km (b), 1 km (c), 500 m (d), 250 m (e) characteristic mesh resolution.

et al., 2019):

$$\boldsymbol{\tau}_b = -C_{\mathrm{W}}\boldsymbol{u}_{\mathrm{b}} \tag{1}$$

$$\boldsymbol{\tau}_b = -\lambda C_{\mathrm{C}}. \left( \frac{\|\boldsymbol{u}_{\mathrm{b}}\|}{\|\boldsymbol{u}_{\mathrm{b}}\| + u_0} \right)^{\frac{1}{m}} \frac{\boldsymbol{u}_{\mathrm{b}}}{\|\boldsymbol{u}_{\mathrm{b}}\|} \tag{2}$$

where $C_{\mathrm{W}}$ and $C_{\mathrm{C}}$ are friction coefficients and $\boldsymbol{u}_{\mathrm{b}}$ is the basal velocity field. This form of regularised Coulomb law, Eq. (2), follows Joughin et al. (2019), which subsumed the potentially nonlinear dependence of effective pressure, $N$, into the friction coefficient, $C_{\mathrm{C}}$. $\lambda$ is used as a scaling factor:

$$\lambda = \begin{cases} 1, & \text{for } h_{af} \geq h_T \\ \frac{h_{af}}{h_T}, & \text{otherwise} \end{cases} \tag{3}$$

with $h_{af}$ the height of ice above flotation and $h_T$ a threshold height. Joughin et al. (2019) demonstrate that the Coulomb friction field has relatively low sensitivity to the choice of parameter $u_0$, and suggest that their parameter setting can be well transferred



for use with general glaciers. We set $u_0 = 300\mathrm{m\,a^{-1}}$, $h_T = 75\mathrm{m}$ for all experiments that use the regularised Coulomb law, following the settings by Joughin et al. (2019) and Hill et al. (2023). $m$ is a positive exponent, often related to the creep exponent $n$ of Glen's law (Glen, 1958) as $m = 1/n$. Here we use $m = 3$, following Hill et al. (2023). We assume a non-linear
isotropic rheology following Glen's flow law (Glen, 1958). For the viscosity, $\eta$, we use

$$\eta = E_\eta^2 \eta_0 \tag{4}$$

Here, $\eta_0$ represents the reference field for $\eta$. It is calculated from a 2-D temperature field, which is obtained by vertically averaging a three-dimensional (3-D) field. The 3-D field is derived from a multi-millennial spin up of the whole Antarctica, utilizing the ice sheet model, SICOPOLIS (Greve et al., 2020; Seroussi et al., 2020). Furthermore, the values for activation
energies and prefactors, essential for computing the temperature-dependent rate factor $A$ in accordance with Glen's flow law, are adopted from Cuffey and Paterson (2010). The term $E_\eta$ in the equation stands for the viscosity enhancement factor, the determination of which will be achieved through inversion processes.

In this study, we invert the basal shear stress and ice viscosity using the refined 1 km resolution mesh (Fig. 2a), with ice velocity observations (Mouginot et al., 2019a, b) as the optimisation target. We employ the linear Weertman law to compute the
basal shear stress in the inversion process. More specifically, we utilise the adjoint inverse method with Tikhonov regularisation, as described in Gillet-Chaulet et al. (2012); Brondex et al. (2019), to invert friction parameter $\beta$ and viscosity enhancement factor $E_\eta$ simultaneously. $\beta$ is given by $C_W = 10^\beta$. The inversion criterion is twofold: to minimise the velocity misfit, and to avoid over-fitting of the inversion solution to non-physical noise in the velocity observation. We introduce three regularisation terms in the total cost function:

$$J_{tot} = J_0 + \lambda_\beta J_{reg\beta} + \lambda_{E_{\eta 1}} J_{regE_{\eta 1}} + \lambda_{E_{\eta 2}} J_{regE_{\eta 2}} \tag{5}$$

The misfit between the magnitudes of simulated ($\boldsymbol{u}$) and observed ($\boldsymbol{u}_{\mathrm{obs}}$) surface velocity is encapsulated in the first cost term $J_0$, which is a discrete sum evaluated directly at every grid node:

$$J_0 = \sum_1^{N_{\mathrm{obs}}} \frac{1}{2} ||\boldsymbol{u} - \boldsymbol{u}_{\mathrm{obs}}||^2 \tag{6}$$

where $N_{\mathrm{obs}}$ is the total number of grid nodes with observations. The terms $J_{reg\beta}$ and $J_{regE_{\eta 1}}$ are implemented to penalise
the first spatial derivatives of $\beta$ and $E_\eta$, respectively. Meanwhile, $J_{regE_{\eta 2}}$ penalises the deviations from the prior (i.e. Glen's flow law; $E_\eta = 1$). The coefficients $\lambda_\beta$, $\lambda_{E_\eta 1}$ and $\lambda_{E_\eta 2}$ are positive regularisation weighting parameters. We determine the optimal combination of these three parameters by conducting an "L-surface" analysis, resulting in $\lambda_\beta = 20000$, $\lambda_{E_\eta 1} = 10000$ and $\lambda_{E_\eta 2} = 0.02$. This "L-surface" analysis represents an innovative aspect of this study and is elaborated upon in Appendix A.

The spatial distributions of the two parameters are shown in Fig. 4a, b, respectively. As shown in Fig. 4c, the velocity difference between inversion result and observations (Mouginot et al., 2019a) was assessed in terms of relative difference. The results indicated that the simulated velocities from the inversions were in good agreement with the observed velocities,



(a)

(b)

(c)

**Figure 4.** The optimised basal resistance parameter $\beta$ (a), viscosity enhancement factor $E_\eta$ (b) and relative surface horizontal velocity discrepancy (c) for the WSB. The relative surface velocity discrepancy is the magnitude of the surface horizontal velocity difference between observations (Mouginot et al., 2019a) and simulations as a fraction of the observations. The three contours (yellow, orange, and red) represent the observed surface velocities of 200, 700, and 1000 m a$^{-1}$, respectively. The white contour in (c) represents the observed surface velocity of 5 m a$^{-1}$. The black line represents the grounding line from BedMachine Antarctica V3 (Morlighem, 2022).





especially in the fast-flow areas where velocities exceed $200 \mathrm{m\,a}^{-1}$ (Fig. 4c). In these fast-flow regions, relative differences are predominantly below 5%. In Fig. 4c, the blue area indicates a high relative velocity discrepancy and corresponds to regions with
very slow flow (mostly below $5 \mathrm{m\,a}^{-1}$). Therefore, it does not present a concern. Such findings underscore that the inversion results can effectively serve as a reliable starting point for subsequent transient experiments. We interpolate the simulated basal friction coefficient $\beta$ and viscosity enhancement factor $E_\eta$ from 1 km resolution grid onto the 250 m, 500 m, and 2 km resolution grids, respectively. These interpolations serve as the starting points for the subsequent historical runs on the four distinct grids (Fig. 3).

## 2.2 Transient simulations

We explore the sensitivity of ice dynamics to the four different GLMPs by conducting a series of transient simulations. After the inversions, we initiate historical runs to smoothly transition the model past an initial adjustment phase in the forward transient simulations (Fig. 2). The historical runs span 20 years, from 1995 to 2015. Then we conduct future runs from 2015 to 2500 (Fig. 2). Each future run is directly paired with a corresponding historical run, maintaining a consistent model con-
figuration throughout. The transient simulations encompass a range of model choices, including two basal friction laws, two climate forcing scenarios, four characteristic mesh resolutions, two ice shelf melt parameterizations (ISMPs) and, as the focus of the study, four GLMPs for the partially floating elements. Each simulation is designated by the naming convention *FL_SSP_RES_ISMP_GLMP*, with the specific components detailed in Table 2.

**Table 2.** Summary of simulation naming convention.

| Name part | Meaning | Possible values |
|---|---|---|
| *FL* | Basal friction law | Weertman or Coulomb |
| *SSP* | Emission scenario of thermal forcing | SSP126 or SSP585 |
| *RES* | Characteristic mesh resolution | 250 m, 500 m, 1 km, 2 km |
| *ISMP* | Ice shelf melt parameterization | NoWCS, WCS75 |
| *GLMP* | Grounding line melt parameterization | NMP, FMP, SEM1, SEM3 |

As the primary focus of this study, we test four GLMPs for partially floating elements, as shown in Fig. 5. We essentially
adopt the parameterization schemes outlined by Seroussi and Morlighem (2018) in an idealised domain. The "full-melt parameterization" (FMP) applies melt across all partially floating elements, irrespective of the grounding line's exact position. Conversely, for the "no-melt parameterization" (NMP), there is no melt applied to any part of these elements. The remaining two schemes employ sub-element parameterizations. In "sub-element melt 1" (SEM1), melt is applied to the entire area of partially floating elements, but its magnitude is reduced based on the fraction area of the floating ice in the element. This
ensures that the total melt over the element is proportionate to the floating ice area. In the "sub-element melt 3" (SEM3), an increased number of 20 integration points are used during the finite element assembly procedure within any partially floating element. We determine the float/ground status for each point and calculate the basal melt rate for the floating points based on its



specific coordinates. It is named SEM3 to differentiate from SEM2 in Seroussi and Morlighem (2018). In essence, our SEM3 aligns with the principles of the sub-element parameterization 3 (SEP3) from Seroussi et al. (2014), which indicate that with

a sufficient number of integration points, the functionality of SEP3 closely mirrors that of the sub-element parameterization 2 (SEP2). Thus, we anticipated that SEM3 in this study will perform similarly to SEM2 as described by Seroussi and Morlighem (2018). For basal friction on the partially floating elements, we consistently adopt SEP3 with 20 integration points for all the transient experiments, following the methods discussed by Seroussi et al. (2014).

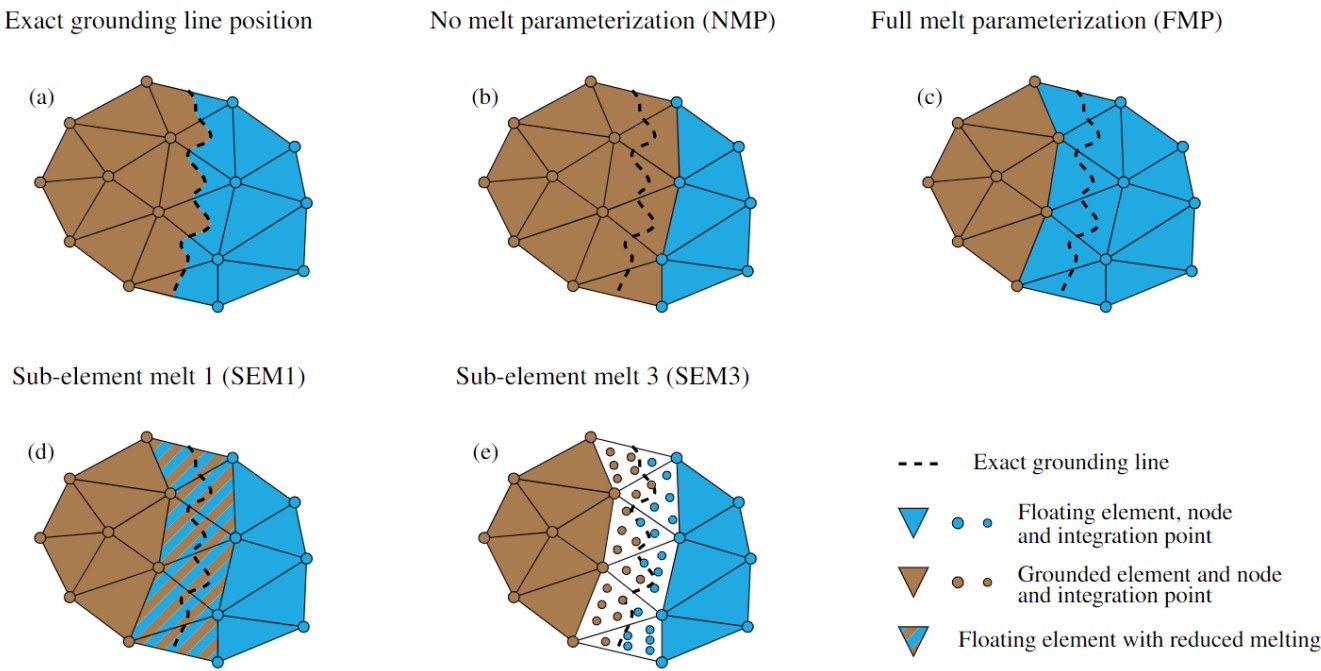

**Figure 5.** Grounding line discretization. Grounding line's exact location (a), no-melt parameterization (NMP, b), full-melt parameterization (FMP, c), sub-element melt parameterization 1 (SEM1, d), and sub-element melt parameterization 3 (SEM3, e). This figure is adapted from Seroussi and Morlighem (2018)

     We impose surface mass balance (SMB) and basal mass balance (BMB) data sourced from the ISMIP6–2300 project

(Seroussi, 2022), based on CMIP6 climated model data, as the forcing. More specifically, the SMB consists of an average value for the reference period, $SMB_{ref}$, and yearly SMB anomalies, $aSMB$:

$$SMB(x,y,t) = SMB_{ref}(x,y) + aSMB(x,y,t) \qquad (7)$$

In this equation, $SMB_{ref}$ represents the temporal average spanning 1995 to 2300 and is derived from MAR simulation products (Agosta et al., 2019). $aSMB$ is calculated based on thermal forcing from climate models, detailed below. Following

the ISMIP6-2300 standard melting parameterization (Seroussi, 2022), the BMB is calculated using a quadratic function of thermal forcing as described by Favier et al. (2019), complemented by a thermal forcing correction suggested by Jourdain





et al. (2020). Building upon this, we produce a revised version whereby the basal melt rate smoothly transitions to zero as it approaches the grounding line:

$$m_s(x,y) = m(x,y)\tanh\left(\frac{H_c}{H_{c0}}\right) \tag{8}$$

where $H_c$ is the water-column thickness beneath the ice shelf, and $H_{c0}$ is a threshold thickness. An empirical value of $H_{c0} = 75\mathrm{m}$ is adopted here, with the justification for this choice detailed in Asay-Davis et al. (2016). This water-column thickness-based scaling is inspired by prior research (e.g. Asay-Davis et al., 2016; Gladstone et al., 2017) and serves as a comparison to Experiment 1 in Seroussi and Morlighem (2018). In the naming convention (Table. 2), this modified ISMP is designated WCS75 (Water Column Scaling with a threshold thickness of 75 m), while the original version is labeled NoWCS (No Water
Column Scaling).

We utilise the thermal forcing provided by the ISMIP6–2300 project (Seroussi, 2022) to determine the BMB and $aSMB$ applied during the transient simulations. Two emission scenarios are adopted in the two CMIP6 models for generating the thermal forcing: one sourced from the CESM2 climate model under SSP5-8.5, and the other from the UKESM1 model under SSP1-2.6. The original forcing data from ISMIP6–2300 project spans the period from 1995 to 2300. Beyond 2300, we
extrapolate the forcing to the year 2500 by randomly sampling values from the 2280 to 2300.

Two basal sliding laws are employed in the sensitivity experiments, the linear Weertman law and the regularised Coulomb law, Eq. (1, 2). The basal friction parameter, $C_W$, for the linear Weertman law is derived directly from inversions. To derive the basal friction parameter, $C_C$, for regularised Coulomb law, we transform the inverted basal friction parameter $\beta$ into $C_C$ by substituting Eq. (1) into Eq. (2):

$$C_\mathrm{C} = \frac{10^\beta}{\lambda}\left(\frac{\|\boldsymbol{u}_\mathrm{b}\| + u_0}{\|\boldsymbol{u}_\mathrm{b}\|}\right)^{\frac{1}{m}}\|\boldsymbol{u}_\mathrm{b}\| \tag{9}$$

This ensures that the basal shear stress remains consistent throughout the conversion process.

## 3 Results

This section presents the results of the future simulations from 2015 to 2500, featuring a comprehensive comparative analysis based upon the time series of two quantitative metrics: total ice mass and total grounding line flux of the model. The analysis
focuses on the high-emission scenario experiments, because we can evaluate the effect of GLMPs best when the grounding line migrates. We also include results from simulations under a low-emission scenario in order to compare. Figures 6 and 7 represent the evolution of total ice mass and the total grounding line flux, respectively, under a high emission scenario (SSP5-8.5) with the application of linear Weertman sliding law. Figures 8 and 9 showcase these variables under the same emission scenario but using a regularised Coulomb sliding law. Figure 10 illustrates the evolution of ice thickness and grounding line
retreat in the future run. Although we have not demonstrated grounding line hysteresis or irreversibility as discussed by Schoof (2007), our projections of rapid grounding line retreat across the retrograde section of the bedrock, compared to the retreat rates across the upsloping bed, strongly indicate that MISI can occur in this region.





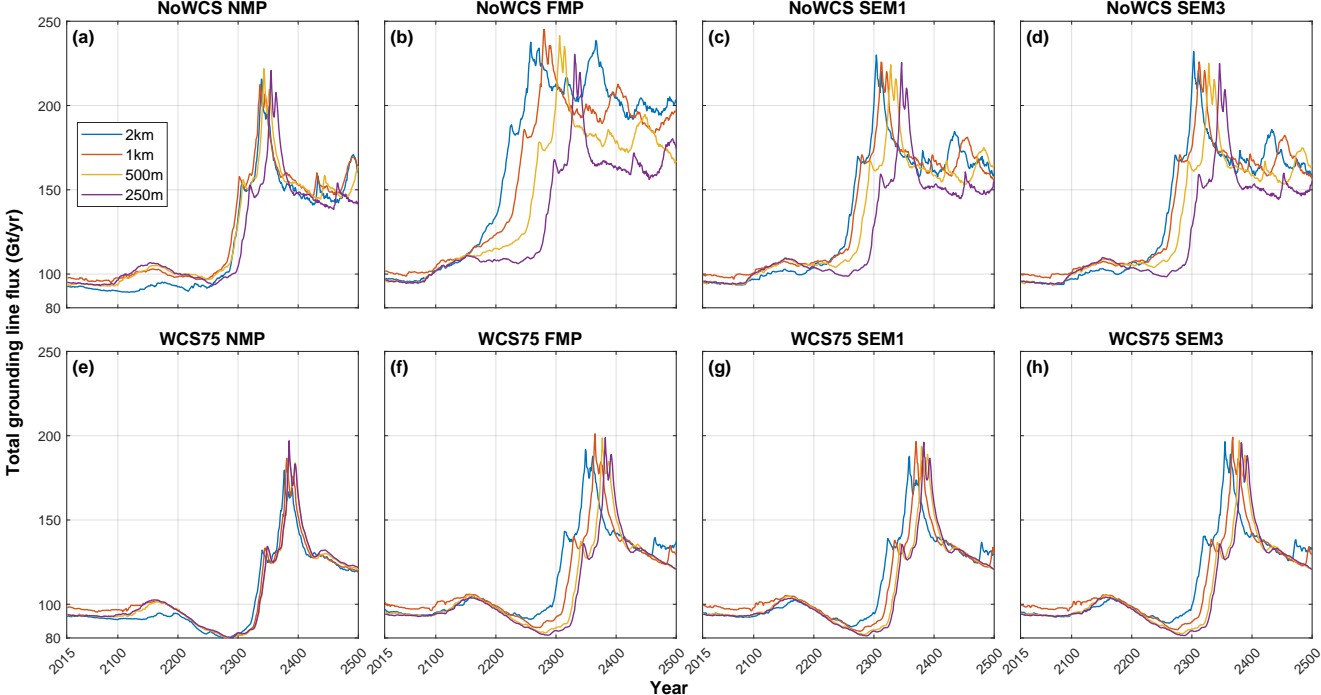

**Figure 6.** Total grounding line flux simulated from 2015 to 2500 under high emission scenario (SSP5-8.5) using a linear Weertman sliding law. The figures are separated by NMP(a, e), FMP(b,f), SEM1 (c,g) and SEM3 (d,h), and two ISMPs: NoWCS (a-d) and WCS75 (f-h). Each plot represents ice flux for the four mesh resolutions: 2 km (blue), 1 km(red), 500 m(yellow) and 250 m (purple).

In the linear Weertman experiments, a majority of the model configurations exhibit a relatively stable grounding line flux over the initial 200-year span (Fig. 6). During this period, the grounding line undergoes a retreat across the comparatively

shallow and flat bed topography, as shown in Fig 10, with persistent ice-shelf thinning mainly caused by the basal melt. This phase is characterised by a stable total ice mass, as shown in Fig. 7. The starting point of the grounding line flux accelerated increase (Fig. 6) signals the tipping point of the MISI, marked by an accelerated retreat of the grounding line into retrograde deep troughs (Fig. 10; after the year 2200), subsequently manifesting as a rapid ice mass loss in Fig. 7. The peak of grounding line flux corresponds to a major rapid retreat of the grounding line within the troughs upstream of Cook glacier (Fig. 10). The

tipping point of the MISI, indicative of a critical transition in ice sheet dynamics, is generally attained around the year 2300 in experiments with water column scaling scheme (Fig. 7 e-f). While for the experiments without the water column scaling (Fig. 7 a-d), the timing of tipping point is significantly advanced. *NoWCS_NMP* reaches the tipping point around 2250 (Fig. 6a); *NoWCS_FMP* around 2150 (Fig. 6b); and both *NoWCS_SEM1* and *NoWCS_SEM3* attain around 2200 (Fig. 6c, d), yielding very similar predictions. Notably, *Weertman_SSP585_2km_NoWCS_FMP* predicts the highest ice mass loss, at $1.04 \times 10^5$ Gt,

doubling that of *Weertman_SSP585_250m_NoWCS_FMP*. This highlights the substantial dependency of the FMP scheme on grid resolution.





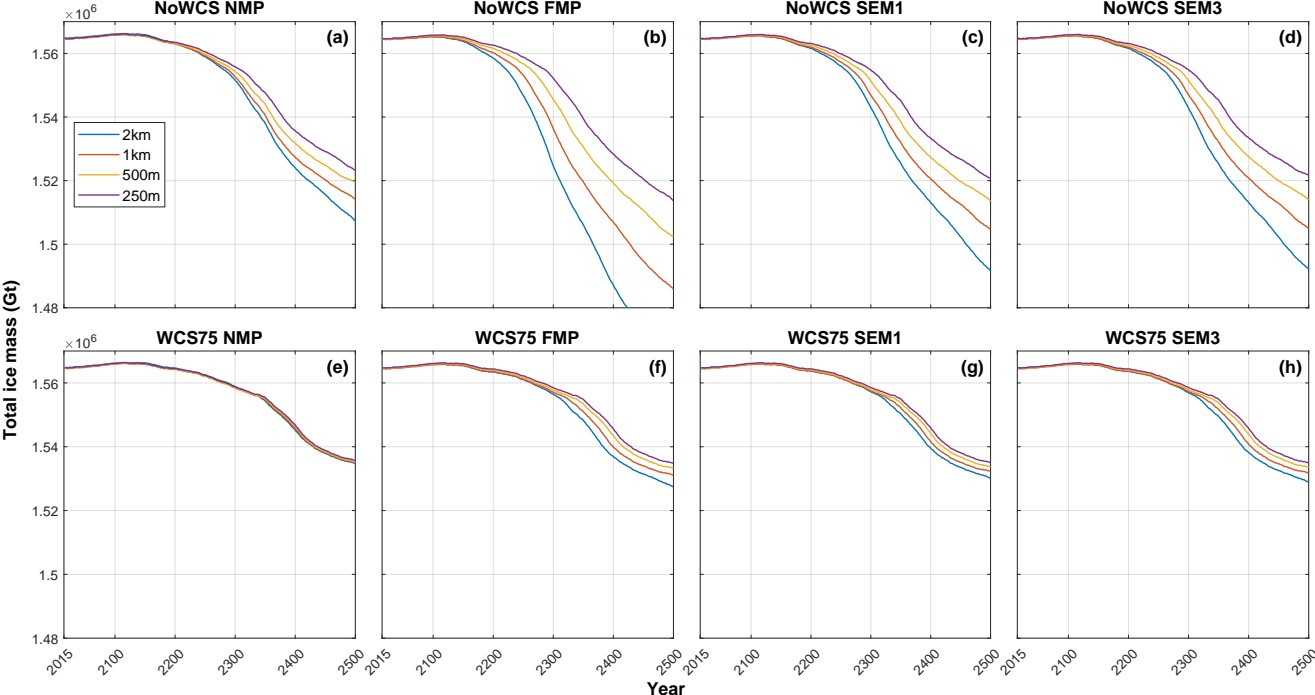

**Figure 7.** Total ice mass simulated from 2015 to 2500 under high emission scenario (SSP5-8.5) using a linear Weertman sliding law. The figures are separated by NMP(a, e), FMP(b,f), SEM1 (c,g) and SEM3 (d,h), and two ISMPs: NoWCS (a-d) and WCS75 (f-h). Each plot represents the ice mass change for the four mesh resolutions: 2 km (blue), 1 km(red), 500 m(yellow) and 250 m (purple).

In the regularised Coulomb experiments, the system is relatively stable for the initial 100 years with NoWCS (Fig. 6a-d) and for around 150 years with WCS75 (Fig. 6e-h), after which the MISI is triggered. A distinguishing feature of the Coulomb experiments is the earlier triggering of the tipping point, compared to the Weertman experiments, and the manifestation of

two distinct peaks in grounding line flux. The two peaks are dominated by the two major rapid retreat of grounding line in troughs upstream of Cook (Fig. 10) and Ninnis glacier respectively. The two peaks are experienced in all experiments without water column scaling scheme (Fig. 6 a-d), while the experiments with water column scaling only experienced the first peak in the last 100 years (Fig. 6 e-f), due to its slower response. The overall magnitude of grounding line flux and ice mass loss of regularised Coulomb experiments are three times greater than those of the linear Weertman experiments. Figure 11 provides

a visual representation of the grounding line position at the year 2500, comparing the four GLMPs under a specific model configuration, *Coulomb_SSP585_1km_WCS75*. The distance between the NMP and FMP grounding lines ranges from 20 to 70 km, while the grounding line locations are consistent between SEM1 and SEM3.

Tables 3 and 4 provide detailed data on total ice mass change from 2015 to 2500 under the linear Weertman and regularised Coulomb laws, respectively. Among the four GLMPs, NMP consistently yields the lowest predictions of ice mass loss; FMP

predicts the highest; SEM1 and SEM3 are intermediate in between. Notably, the Weertman and Coulomb experiments reveal



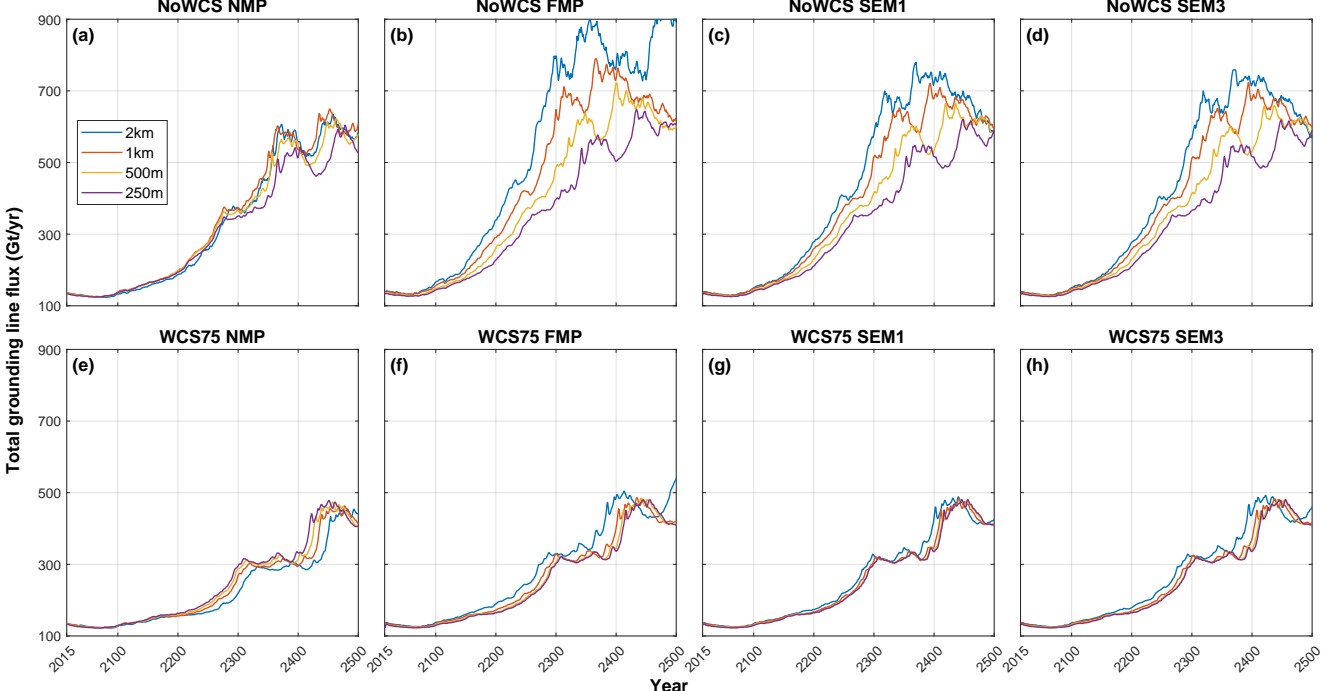

**Figure 8.** Total grounding line flux simulated from 2015 to 2500 under high emission scenario (SSP5-8.5) using regularised Coulomb sliding law. The figures are separated by NMP(a, e), FMP(b,f), SEM1 (c,g) and SEM3 (d,h), and two ISMPs: NoWCS (a-d) and WCS75 (f-h). Each plot represents ice flux for the four mesh resolutions: 2 km (blue), 1 km(red), 500 m(yellow) and 250 m (purple)

different yet internally consistent patterns of total grounding line flux. The resolution dependence on the different parameterisations for partially grounded elements is comparable for both linear and the regularised Coulomb sliding laws, with the exception that coarse resolution underestimates mass loss only in the case of WCS75 NMP Coulomb sliding. The choice of GLMPs exerts a significant impact on both the timing of the tipping point triggered and the cumulative magnitude of ice mass

loss at coarse resolution, while the incorporation of a water column scaling scheme can significantly diminish the discrepancies caused by different GLMPs and mesh resolutions.

Regarding the low emission experiments, we have opted to only present the results at 1 km grid resolution and using only the regularised Coulomb sliding law (Fig. 12), as it did not exhibit notable distinctiveness as compared with the results of high emission experiments. Without the water column scaling procedure, the system exhibits a continuous, albeit slight loss of ice

during the entire future simulation (Fig. 12 a), and there is a substantial discrepancy in the total ice mass change (Fig. 12 a) and total grounding line flux (Fig. 12 b) across different GLMPs. With water column scaling, the system experiences a discernible ice mass loss in the first 50 years; however, it subsequently stabilises (Fig. 12 c). The discrepancy is substantially reduced when the water column scaling is applied (Fig. 12 c, d), indicating a mitigation of the impact of melt scheme selections. In general,



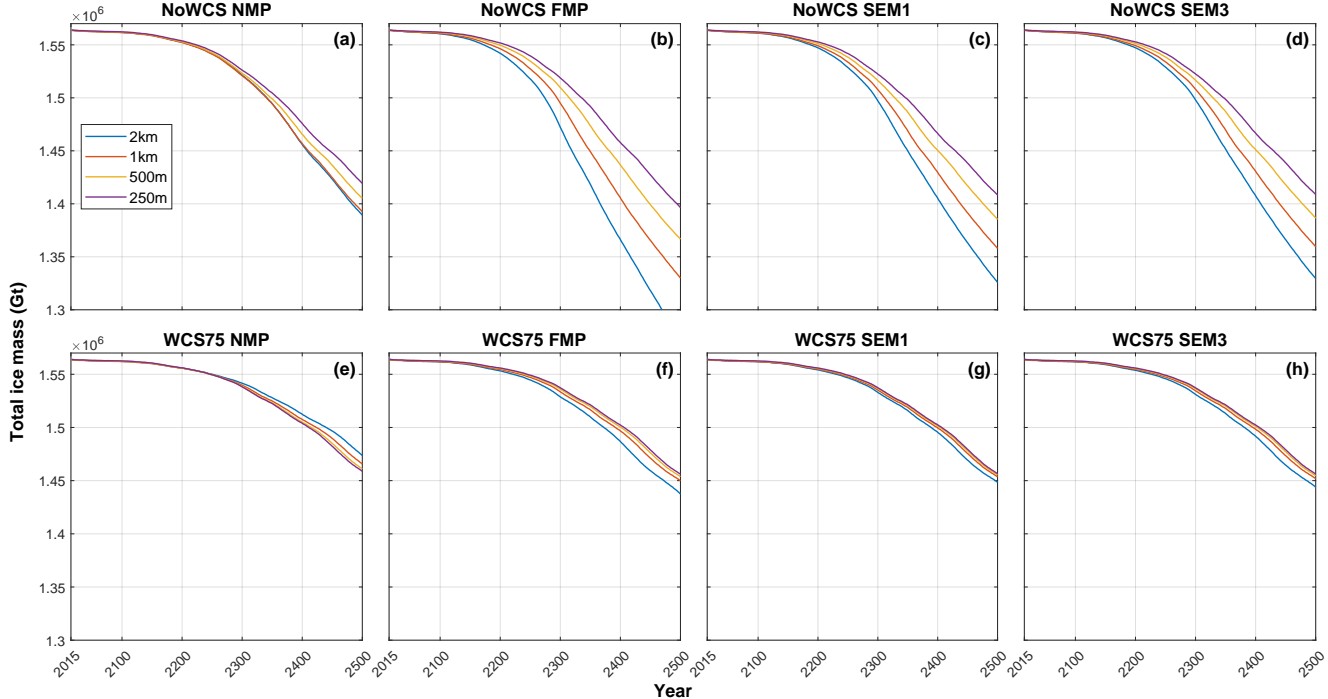

**Figure 9.** Total ice mass simulated from 2015 to 2500 under high emission scenario (SSP5-8.5) using regularised Coulomb sliding law. The figures are separated by NMP(a, e), FMP(b,f), SEM1 (c,g) and SEM3 (d,h), and two ISMPs: NoWCS (a-d) and WCS75 (f-h). Each plot represents the ice mass change for the four mesh resolutions: 2 km (blue), 1 km(red), 500 m(yellow) and 250 m (purple).

under a low emission scenario, the predicted ice mass loss is less sensitive to the choice of GLMPs and mesh resolution in
comparison to high emission scenarios.



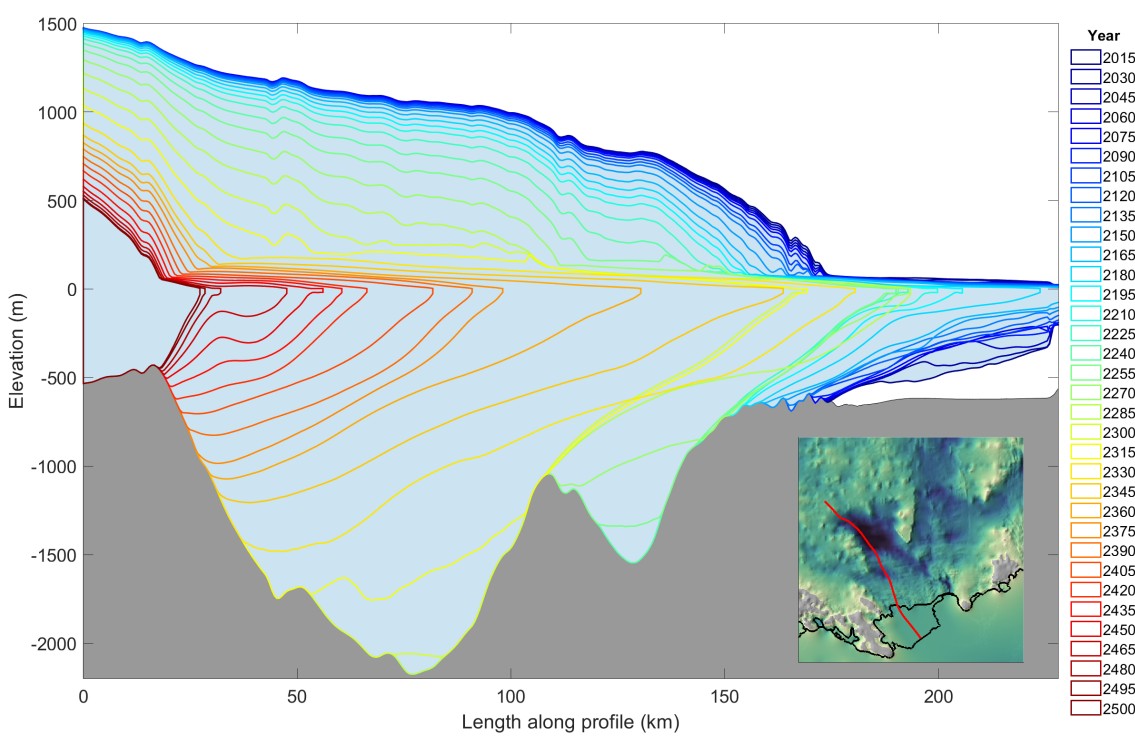

**Figure 10.** The evolution of ice thickness along a characteristic flowline on the Cook glacier, as projected in the future run *Coulomb_SSP585_500m_WCS75_SEM3* for illustration. The rainbow coloured outlines represent the time series progression of ice thickness in the future run. The inset shows the location of the flowline in red. For better visual presentation, ice at the front with thickness less than 20 m are not shown.



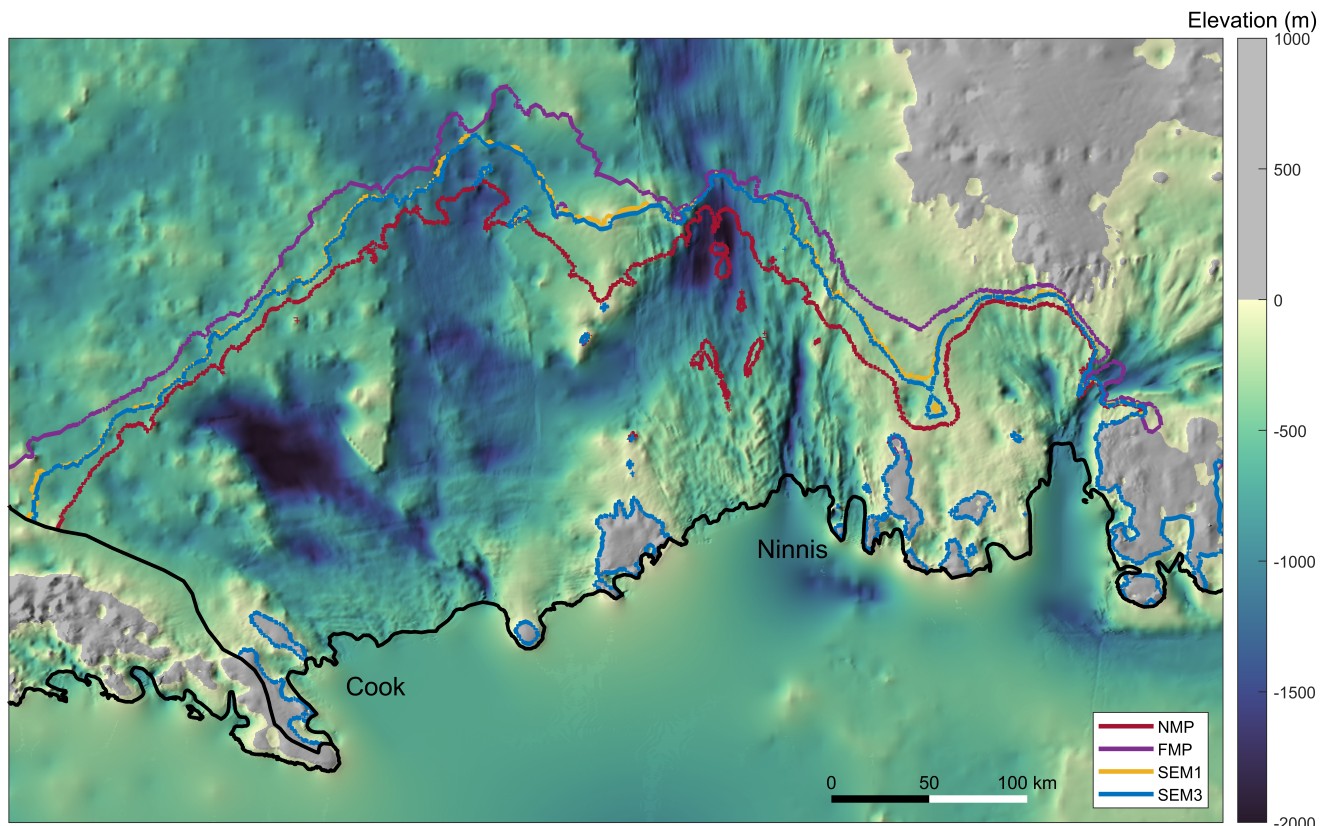

**Figure 11.** The simulated grounding line at the year 2500 with NMP (red), FMP (purple), SEM1 (yellow), SEM3 (blue) are presented on the bed topography map. They all are under high emission scenario (SSP5-8.5), without the water column scaling scheme, using regularised Coulomb law at 1 km grid resolution. The grounding line of SEM1 and SEM3 are largely overlap. The grounding line of all four GLMPs overlaps around ice rises, covered by blue grounding lines. The position of the Cook and Ninnis glacier are marked.



**Table 3.** Change in total ice mass (Gt) under high emission scenario (SSP5-8.5) from 2015 to 2500 with linear Weertman friction law for NoWCS (a) and WCS75 (b)

| (a) | Melt parameterization (NoWCS) | | | |
|---|---|---|---|---|
| Resolution | NMP | FMP | SEM1 | SEM3 |
| 2 km | -57799 | -103715 | -73319 | -72807 |
| 1 km | -50636 | -78809 | -60042 | -59737 |
| 500 m | -45245 | -62448 | -51152 | -50857 |
| 250 m | -41479 | -51510 | -44329 | -43204 |
| (b) | Melt parameterization (WCS75) | | | |
| Resolution | NMP | FMP | SEM1 | SEM3 |
| 2 km | -30194 | -37396 | -34794 | -36052 |
| 1 km | -29637 | -33530 | -32338 | -32858 |
| 500 m | -29497 | -31396 | -30949 | -31130 |
| 250 m | -29179 | -30055 | -29851 | -29946 |

**Table 4.** Change in total ice mass (Gt) under high emission scenario (SSP5-8.5) from 2015 to 2500 with regularised Coulomb friction law for NoWCS (a) and WCS75 (b)

| (a) | Melt parameterization (NoWCS) | | | |
|---|---|---|---|---|
| Resolution | NMP | FMP | SEM1 | SEM3 |
| 2 km | -175276 | -299921 | -238787 | -235084 |
| 1 km | -171745 | -234377 | -206355 | -204773 |
| 500 m | -159045 | -197891 | - 179357 | -178103 |
| 250 m | -145699 | -168594 | -156673 | -156217 |
| (b) | Melt parameterization (WCS75) | | | |
| Resolution | NMP | FMP | SEM1 | SEM3 |
| 2 km | -90755 | -126957 | -115825 | -120388 |
| 1 km | -98431 | -114097 | -110440 | -112088 |
| 500 m | -103380 | -110595 | -109121 | -109743 |
| 250 m | -105871 | -109585 | -108194 | -108442 |





**Figure 12.** The evolution of total ice mass (a, c) and total grounding line flux (b, d) from 2015 to 2500 with four GLMPs. The results represent the experiments using regularised Coulomb sliding law under the low emission scenario (SSP1-2.6) at 1 km mesh resolution without (a, b) and with (c, d) water column scaling procedure.



## 4 Discussion

In Figure 13, we show the convergence of simulated ice mass loss with mesh resolution for different ISMPs and sliding laws. Our model, which simulates the real-world domain of the WSB, demonstrates a consistent convergence pattern with the idealised glacier model study by Seroussi and Morlighem (2018), showcasing a commendable level of agreement between the
two ice sheet models, Elmer/Ice and ISSM.

**Figure 13.** Convergence of total ice mass loss from 2015 to 2500 as a function of mesh resolution with the four GLMPs, NMP (blue), FMP(green), SEM1(orange), and SEM2(red). The results represent the experiments under the high emission scenario (SSP5-8.5) for the Weertman (a, b) and Coulomb (c, d) sliding law with (b, d) and without (a, c) water column scaling procedure.





Among the four GLMPs, NMP tends to converge more rapidly with resolution in most cases, which is consistent with the findings of Seroussi and Morlighem (2018); Yu et al. (2018). Our model results reveal a trend across all scenarios where ice mass loss diminishes as mesh resolution increases, except for the NMP scheme with the Coulomb law and water column scaling (Fig. 13d; Coulomb_SSP585_WCS75_NMP). In this scenario, the simulated ice mass loss actually increases with finer mesh
resolutions. This result aligns with the simulation results from previous studies (Seroussi and Morlighem, 2018; Leguy et al., 2021; Berends et al., 2023). A plausible explanation lies in the methodology of NMP, which, by definition, underestimates melt in partially grounded elements. As resolution becomes finer and elements become smaller, the area of no melting decreases, resulting in an increase in the area of melting close to the grounding line. However, this does not explain why NMP still overestimates mass loss in other cases, as resolution dependence exists not only due to choice of GLMP, but also due to the
sub-element parameterisation of basal drag near the grounding line. The current study does not investigate impacts of basal drag on convergence with resolution, which has been more extensively studied, but the effects are present and not easily separated from the effects of melt parameterisation. The cumulative impact of parameterisations on both basal drag and grounding line melt is likely what determines convergence. Caution must be exercised regarding the potential for NMP to systematically under-represent melt at the grounding line and thus underestimate ice mass loss at coarse grid resolutions.

Conversely, FMP, by definition, overestimates melt in partially grounded elements, and our simulations using FMP always overestimate ice mass loss. In the experiments without water column scaling, the total ice mass loss simulated at a 2km resolution is approximately double that simulated at a 250 m resolution (Fig. 13a, c). We notice that the ice sheet modelling community has largely moved away from the FMP scheme. We align with this perspective and concur with prior recommendations (Leguy et al., 2021; Seroussi and Morlighem, 2018) that the FMP scheme should be avoided under all circumstances.

Whilst FMP and NMP by definition always overestimate and underestimate melt in partially grounded elements, SEM1 and SEM3 are expected to fall in between and therefore give a more accurate estimation of melt in partially grounded elements. However, this does not translate into better convergence with resolution, with most simulations from both the current study and the work of Seroussi and Morlighem (2018) showing significant overestimation of mass loss and grounding line retreat when using SEM1 or SEM3 at coarse resolutions. This issue likely stems from fundamentally under-resolving the problem (i.e., the
model's spatial resolution is insufficient to accurately capture and represent the dynamics at the grounding line). Although SEM1 and SEM3 provide a more viable average melt rate over partially grounded elements, the fact that they can cause thinning directly at "grounded" nodes (Fig. 5) leads to ice detachment that would not occur with a fully resolved model (i.e., one with infinitely small elements, which is unachievable in practice). Consequently, this results in an overestimation of mass loss and grounding line retreat. A more thorough handling of the partially grounded elements might be to implement runtime
adaptivity with a specific focus on the grounding line itself, either by splitting partially grounded elements or by implementing a moving mesh that tracks grounding line movement (Goldberg et al., 2009), but these approaches are beyond the scope of the current study.

The results of SEM1 and SEM3 are consistently falling in between FMP and NMP results (Fig. 11-13). The two sub-element GLMPs give almost identical results without water column scaling, which is similar to findings of the basal friction
parameterizations at the grounding line (Seroussi et al., 2014). Yet, with water column scaling, SEM1 and SEM3 diverge





slightly, with SEM1 showing better convergence with resolution than SEM3. The SEM1 scheme shows the best convergence in the scenario with the Coulomb law and water column scaling. This appears contrary to the recommendation by Seroussi and Morlighem (2018) against the use of SEM due to its overestimation of retreat of the grounding line. While NMP usually shows better convergence, SEM1 appears to outperform in specific scenarios, offering superior convergence.

In the vicinity of the grounding line, high melt rates essentially worsen the convergence with resolution and exacerbate the result discrepancies observed across all four GLMPs. This phenomenon is reflected in different aspects of the experimental results. Firstly, the water column scaling procedure significantly improves the convergence, and reduces the disparities among the GLMPs (Fig. 7, 9). This is because when water column scaling is applied, the melt rates are significantly reduced near grounding line, thereby minimising the divergences represented by different GLMPs. Secondly, under a high emission scenario,
the predicted ice mass loss is more sensitive to the choice of GLMPs and mesh resolution in comparison to low emission scenarios. In other words, the difference in simulated ice mass loss caused by the various GLMPs are significantly amplified under high emission scenario, as has been demonstrated by Arthern and Williams (2017) using a model of Pine Island and Thwaites glacier.

        Numerical simulation methods and grid type significantly influence the performance of GLMPs. Consistent with previous
model studies (e.g. Seroussi and Morlighem, 2018; Yu et al., 2018), our research employs the finite-element method with an unstructured triangular grid, and concludes that NMP always demonstrates superior convergence with resolution compared to FMP and usually compared to SEM. Notably, a model study (Arthern and Williams, 2017) employing a finite-volume method and a wavelet-based adaptive grid demonstrated significant impact of sub-grid GLMPs on ice mass loss predictions, echoing our findings. However, respective studies by Leguy et al. (2021) and Berends et al. (2023), utilising finite differences
and a regular square grid, report contrary findings. Due to the distinct mechanism of the model implementation, the GLMPs they used differ from the four explored in our study. In addition to the NMP scheme (identical to ours), they incorporate a Partial Melt Parameterization (PMP; comparable to our SEM1) and a Flotation Condition Melt Parameterization (FCMP). Both FCMP and PMP outperform NMP in terms of convergence with resolution (Leguy et al., 2021; Berends et al., 2023). Their agreement implies that for models using finite differences and regular grids, applying melt parameterizations to partially
floating cells could be more advantageous. Thus, it is crucial to reevaluate the performance of various GLMPs within specific model contexts.

        Modelling studies emphasise the necessity of including significant melting processes within the grounding zone to replicate the observed retreat patterns (Goldberg et al., 2019; Lilien et al., 2019). Further, satellite observations indicate pronounced melt rates at the grounding lines in both West Antarctica (Shean et al., 2019) and Greenland (Ciracì et al., 2023). Drawing from
the observations, Ciracì et al. (2023) recommend that ice sheet models adopt GLMPs that include melting at and upstream of the grounding line. We acknowledge the scientific rationale behind this suggestion; however, it may not directly translate to the parameterization strategies for the partially floating elements in ice sheet models. It is crucial to distinguish between the role of ISMPs and the specific function of GLMPs. We suggest that the ISMPs should reflect our current best understanding of ice-ocean interactions near the grounding line. Meanwhile, the design of GLMPs ought to prioritise model self-consistency
and minimal resolution dependency.





The melting mechanism and the precise rates at the grounding line are still not well understood (Goldberg et al., 2019). Our NoWCS and WCS75 schemes encapsulate the divergent perspectives currently debated: one posits that the maximum melt rate occurs right at the grounding line, where the ice draft is deepest (e.g. Ciracì et al., 2023; Shean et al., 2019), whereas ocean modelling studies suggest that the melt rate peaks downstream of the grounding line and diminish to zero towards the ground-

ing line (e.g. Burgard et al., 2022; Slater et al., 2020). A recent theoretical study suggest the possibility of melting extending upstream of the grounding line if warm salty seawater intrudes under the ice sheet (e.g. Robel et al., 2022). However, observations at Thwaites Glacier show melting at the grounding line there is suppressed by low current speeds and strong density stratification in the ice–ocean boundary layer (Davis et al., 2023). Moreover, melting processes and ice-ocean interactions at the grounding line likely differ among ice shelves due to variations in bathymetry, water mass characteristics, and circulation

beneath ice shelf cavities. To elucidate the melting mechanisms at play, there is a critical need for more extensive observations of melt rates in the vicinity of grounding lines.

The extensive exploration of model settings in this study underscores the significant uncertainties inherent in ice sheet modelling predictions. Utilising the Coulomb sliding law, which is broadly considered superior, our analysis under the high emission scenario of SSP5-8.5 suggests that the tipping point (onset of the MISI; marked by a rapid increase in grounding

line flux, as shown in Fig. 8) is anticipated to occur in WSB between 2200 and 2300. After the tipping point, the grounding line retreats 110 km across the unstable retrograde bedrock in 100 years (as illustrated in Fig. 10). The grounding line flux consequently increases by a factor of 2.5, from 200 to 500 $\mathrm{Gt\,a^{-1}}$ (Fig. 8). In this context, our simulations project an ice mass loss within the WSB in this scenario to range from 0.26 to $0.42 \times 10^5$ Gt by the year 2300. This corresponds to a mass above flotation of 0.21 to $0.33 \times 10^5$ Gt, equivalent to 0.06 to 0.09 m of global sea level rise. By 2500, the projected ice mass loss

extends from 1.05 to $1.57 \times 10^5$ Gt, corresponding to a mass above flotation of 0.84 to $1.25 \times 10^5$ Gt, equivalent to a global sea level rise of 0.23 to 0.34 m, assuming the extension of the final two decades' forcing from 2300. At a mesh resolution of 1 km, which is commonly employed in ice sheet modellings, our model shows a change from NMP to SEM would induce a 15% to 20% increase in projected ice mass loss. Moreover, at a 1 km resolution, SEM could overestimate mass loss by up to 40% compared to our finest mesh resolution of 250 m, whereas NMP might overestimate it by up to 25% relative to the 250

m mesh, with specific overestimations dependent on the model configurations (Fig. 13). These results provide a foundation for further detailed quantitative predictions and the examination of ice sheet dynamics in future stages of our ongoing research.

In our comparative analysis, both SEM and NMP schemes outperform FMP. As discussed earlier, SEM and NMP exhibit distinct advantages, each conducive to certain modelling contexts. The suitability of GLMPs is contingent upon the specific model and circumstances in question. The alignment between the results from idealised model simulations (Seroussi and

Morlighem, 2018) and our comprehensive real-domain model experiments support the validity of a two-phased experiment process: one could firstly evaluate the performance of various GLMPs based on a cost-effective, idealised small ice flow model (e.g. MISMIP+; Cornford et al., 2020) and then inform subsequent applications to more complex real-world domains. For the future explorations, mesh adaptation and re-segmentation at the sub-element scale during runtime would be a promising direction for more accurately representing basal friction and melting at the grounding line.



## 5   Conclusions

In this study, we explored the sensitivity of future projections of the Wilkes Subglacial Basin (WSB) ice sheet to Grounding Line Melt Parameterizations (GLMPs) for the partially floating elements separating the grounded ice and the floating shelf. The study is conducted through a series of model simulations for the WSB spanning from 2015 to 2500. These simulations test the performance of four GLMPs under various model configurations, incorporating two basal friction laws, two thermal forcing scenarios, four mesh resolutions, and two Ice Shelf Melt Parameterizations (ISMPs). Drawing from our best model results, the tipping point, onset of the MISI, is projected to occur between 2200 and 2300 in the WSB under the high emission scenario of SSP5-8.5, while the ice sheet system is expected to remain a quasi-steady state under the low emission scenario of SSP1-2.6. Under SSP5-8.5, our simulations suggest that the loss of ice from the WSB could contribute between 0.06 to 0.09 m to global sea level rise by 2300, while following the onset of MISI, this contribution could increase to between 0.23 to 0.34 m by 2500.

Our findings indicate that the GLMPs significantly affect both the timing of the tipping point triggered and the overall magnitude of ice mass loss. At a resolution considered high and commonly employed in ice sheet models (i.e., 1km), numerical errors due to inadequate convergence can lead to an overestimation of mass loss by up to 40% when compared to our finest mesh resolution of 250 m. This magnitude of overestimation is comparable to the impact of variations in basal friction parameterizations at the grounding line. In the vicinity of the grounding line, high melt rates notably impair convergence with resolution and amplify the result discrepancies among the four GLMPs. This underscores the critical importance of not only knowing what melt rates are from an observational perspective, but also choosing the appropriate melt parameterization in such scenarios.

Overall, both SEM and NMP schemes outperform FMP in terms of mesh resolution convergence, with each exhibiting varying degrees of superiority over the other. The NMP scheme, in most scenarios, yields superior convergence of results, but may systematically underestimate grounding line retreat and ice mass loss. Conversely, the SEM exhibited better convergence in the scenario with the Coulomb sliding law and water column scaling procedure. The SEM technically can more accurately represent the amount of melting in partially grounded elements and may capture some grounding zone-like transitional behaviours, but it risks overestimating ice mass loss. As in prior studies (Seroussi and Morlighem, 2018; Leguy et al., 2021), we advise against the FMP under all circumstances, due to its poor convergence and substantial overestimation of ice mass loss.

Our research suggests that there is currently no universally optimal melt scheme that suits all circumstances; the choice between NMP and SEM should be re-evaluated in their specific model contexts. Looking ahead, mesh adaptation and re-segmentation at the sub-element scale during runtime emerge as promising avenues for more accurately representing basal friction and melting at the grounding line. Idealised models, such as MISMIP+ (Cornford et al., 2020), provide valuable insights for selecting GLMPs in more complex real-world domains. These improvements are critical to enhancing the accuracy of future predictions of ice mass loss and global sea level rise.



*Code and data availability.* All model simulations are implemented using Elmer/Ice Version: 9.0 ( ReV: bf10af7; doi.org/10.5281/zenodo.7892181) with the code available at https://github.com/ElmerCSC/elmerfem.git (Gagliardini et al., 2013). Mesh and implementation scripts for the model are available at github link. Detailed output data for the model are available upon request to YW.

**Appendix A: "L-surface" analysis**

In our cost function (Eq.5), we introduce three undetermined regularisation parameters. Consequently, the conventional L-curve analysis is insufficient for our purposes, leading us to propose a more comprehensive "L-surface" analysis.

Throughout our analytical experiments, we adopt an empirical value of 0.02 for $\lambda_{E_\eta 2}$, as sensitivity experiments indicate that the inversion results are relatively insensitive to variations $\lambda_{E_\eta 2}$ (as corroborated through personal communication with Fabien

Gillet-Chaulet). To optimise the remaining regularisation parameters $\lambda_\beta$ and $\lambda_{E_\eta 1}$, we undertake a systematic exploration of their feasible value combinations. As an initial step in our L-surface analysis, we conduct preliminary experiments to identify appropriate alternative values for these parameters. Specifically, we select 9 test values for $\lambda_\beta$ and 10 for $\lambda_{E_\eta 1}$. Pairwise combinations of these test values yield 90 distinct parameter sets for subsequent inversion experiments. The results of these experiments are presented in a 3-D visualisation, as depicted in Figure A1.

To identify the optimal combination of $\lambda_\beta$ and $\lambda_{E_\eta 1}$, we employ a metric defined as the relative distance, $D_{rel}$, from each point to the origin in the 3-D coordinate system:

$$D_{rel} = \sqrt{(\frac{J_0}{\max(J_0)})^2 + (\frac{J_{reg\beta}}{\max(J_{reg\beta})})^2 + (\frac{J_{regE_\eta 1}}{\max(J_{regE_\eta 1})})^2} \qquad (A1)$$

The point corresponding to the smallest $D_{rel}$ value is deemed to represent the most favourable combination of $\lambda_\beta$ and $\lambda_{E_\eta 1}$, marked as red star in Fig. A1. Through the "L-surface" analysis, we determine the optimal parameter set to be $\lambda_\beta = 20000$,

$\lambda_{E_\eta 1} = 10000$ and $\lambda_{E_\eta 2} = 0.02$.

*Author contributions.* YW, CZ and RG designed the experiments together. YW, CZ, RG, and TZ implemented the model simulations. YW processed, analysed and visualised the simulation results. YW drafted the paper. All authors contributed to the refinement of the experiments, the interpretation of the results and the final paper.

*Competing interests.* The co-author, Ben Galton-Fenzi, is a member of the editorial board of The Cryosphere

*Acknowledgements.* Yu Wang, Chen Zhao and Ben Galton-Fenzi received grant funding from the Australian Government as part of the Antarctic Science Collaboration Initiative program (ASCI000002). Chen Zhao is the recipient of an Australian Research Council Discovery Early Career Researcher Award (project number DE240100267) funded by the Australian Government. This research/project was undertaken




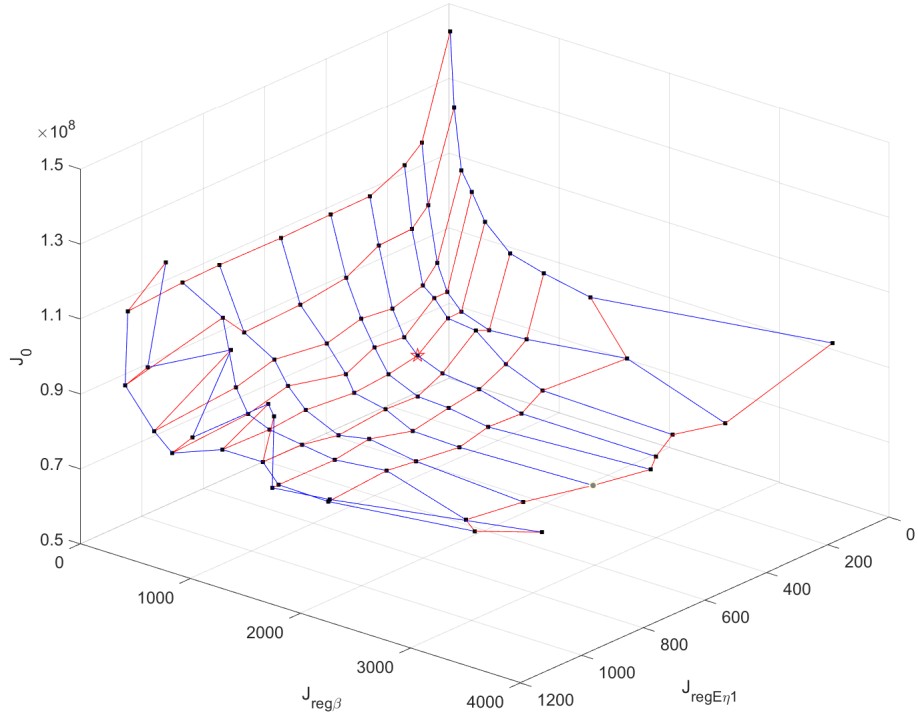

**Figure A1.** "L-surface": black points represented the results from 90 parameter sets. Points connected by the red line correspond to the same $\lambda_\beta$, and points connected by the blue lines correspond the same $\lambda_{E_\eta 1}$. The nine alternative values for $\lambda_\beta$ are 2000, 5000, 10000, 20000, 50000, 100000, 200000, 500000, 1000000. The ten alternative values for $\lambda_{E_\eta 1}$ are 10, 100, 1000, 2000, 5000, 10000, 20000, 50000, 100000, 1000000.

with the assistance of resources and services from the National Computational Infrastructure (NCI), which is supported by the Australian Government. Rupert Gladstone and Thomas Zwinger were supported by Academy of Finland grant nos. 322430 and 355572, and wish to acknowledge CSC – IT Centre for Science, Finland, for computational resources. Rupert Gladstone was also supported by the Finnish Ministry of Education and Culture and CSC - IT Center for Science (Decision diary number OKM/10/524/2022).



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
