# Peer review of "Sensitivity of Future Evolution of the Wilkes Subglacial Basin Ice Sheet to Grounding Line Melt Parameterizations"

_EGUsphere, 2024_

## Author Response (AR1)

**Please note our responses are in bold.**

Review of Wang et al. (2024): "Sensitivity of Future Projections of the Wilkes Subglacial Basin Ice Sheet to Grounding Line Melt Parameterizations"

By Tijn Berends

Ice-sheet models currently dominate the uncertainty in projections of future sea-level rise. A significant part of this model uncertainty stems from the way the discontinuity of the basal melt rate at the grounding line is treated in spatially discrete models. In this study, the authors present experiments with the Elmer/ice model, where they study the future retreat of the Antarctic ice sheet in the Wilkes basin, using different model resolutions, different sliding laws, and different ways to parameterise sub-shelf melt near the grounding line.

In general, I find this a very well-written paper. The experiments are well-defined, the results are presented clearly and concisely, and the conclusions are well-supported by the evidence presented. Having published a similar paper myself quite recently, I am glad to see that someone now did a better job of it! I do have a few comments that I think should be addressed before publishing, but as none of these should lead to additional experiments, I think these warrant "minor revisions" only.

Major comments

An interesting new feature you present is the "water-column scaling", which is based on plume modelling studies (although you also cite observation-based papers that suggest that significant melting still occurs at, and even upstream of, the grounding line at several locations in Greenland and Antarctica). However, the way it is presented now in the results is, in my view, slightly confusing. The GLMPs exist in the realm of "model implementation", i.e. different ways to discretise and solve the same physics. The WCS scheme, however, represents different physics, altering the mass budget of the ice sheet regardless of the choice of model implementation. I think it is important to make this distinction, especially since in the discussion section, you discuss the implications of significant melt upstream of the grounding line.

**Thank you for your insightful suggestion. We completely agree with your understanding and have added the following statement in the Methods section to clarify the distinction between ISMPs and GLMPs: "It is important to distinguish between the roles of ISMPs and GLMPs. ISMPs essentially represents two distinct physical assumptions regarding the melt rate around the grounding line, whereas GLMPs represent different parameterized implementations of the model."**

Both in the abstract and in several places throughout the manuscript, you state that you consider your simulations as actual projections of future mass loss. While I don't object

to this per se, I wonder if this is a good idea. If you want to do this, you will need to provide a lot more information about your experimental set-up (see my technical comments below, about the historical experiments, atmosphere & ocean forcing, etc.). This will add a bunch of extra text to your paper, which won't do anything for the main story (concerning the GLMPs). I also wonder what the added value is of projections of a single ice-sheet basin, especially one that only contributes 30 cm of sea-level rise at most over the course of nearly five centuries. The main users of ice-sheet projections typically want the mass loss of the entire ice sheet, so they can calculate sea-level rise. As the main focus of your paper seems to be model-oriented, which I think is valuable enough by itself, I advise you to consider removing the label of "projections" from your simulations.

**Prior to submitting this manuscript, the authors had differing opinions on whether to include quantitative projections. We fully understand your concerns regarding this matter. After thorough discussion and reconsideration, we have decided to remove the content related to quantitative projections from the abstract and conclusion sections to enhance the focus of the main narrative. Additionally, we have revised the title to "Sensitivity of Future Evolution of the Wilkes Subglacial Basin Ice Sheet to Grounding Line Melt Parameterizations."**

Technical comments

Abstarct: the water column scaling is not mentioned in the abstract

**Thank you for pointing this out. We have now included this in the abstract. It reads: "This study investigates future ice sheet dynamics in the WSB with respect to four GLMPs under both the upper and lower bounds of climate warming scenarios from the present to 2500, with different model resolutions, ice shelf melt parameterizations (ISMPs) and choices of sliding relationships"**

L 54-55 "Modelling studies suggest that ice sheet models are more sensitive to melt rates near the grounding line than to cavity-integrated melt rates beneath ice shelves" What about Joughin et al., 2021: "we find only minor sensitivity to melt distribution (<6%), with a linear dependence of ice loss on the total melt"

**We acknowledge the controversy surrounding this topic. However, the discussion of these contradictory conclusions diverges from the original intention of the paragraph. To convey this more rigorously, we have revised the statement to: "Modelling studies suggest that ice sheet models may be more sensitive to melt rates near the grounding line than to cavity-integrated melt rates beneath ice shelves."**

L 59-60 "...due to the discretisation of the ice sheet model, there inevitably exist grid cells or elements at the grounding line where ice is partially grounded and partially floating" In fixed-grid models, yes. Maybe not something to discuss here, but should we eventually move to moving-grid models?

**We agree that future ice sheet models will likely implement a moving-grid scheme. While challenges such as repeated re-interpolation, re-projection, and dynamic resolution adjustment remain, we have revised the statement for clarity: "However, due to the discretization of the general fixed-grid ice sheet model, there inevitably exist grid cells or elements at the grounding line where ice is partially grounded and partially floating."**

L 88 "...such as the Shallow Shelf Approximation we use here" I've always wondered what the impact of this choice is when combined with an inversion method. Near the grounding line it's probably fine, but further inland there should at present be at least some vertical shearing going on. The SSA neglects this, so the inversion must by necessity overestimate the basal slipperiness to compensate. Near the end of your projections, the grounding line might retreat into this area of overestimated slipperiness, artificially amplifying the retreat (see also the "compensating errors" in Berends et al., 2022 - https://tc.copernicus.org/articles/17/1585/2023/tc-17-1585-2023.pdf). Not something to investigate here, obviously, but maybe something to mention.

**We agree that the SSA might lead to an overestimation of retreat on centennial timescales. However, since our paper no longer aims to provide quantitative predictions, the limitations of the SSA are beyond the scope of our discussion. Including this topic would dilute the focus of our paper. Thus, we have decided not to incorporate this topic into our discussion.**

L 98 "The locations of calving front and inland boundary are held fixed throughout the simulations" This needs some elaboration. Is the ice not allowed to advance beyond that front but allowed to retreat within it, or is the front really fixed? If so, how? Do you apply a minimum ice thickness to maintain a thin shelf within the observed front?

**Thank you for the suggestion. The ice is not allowed to advance beyond the front but is allowed to retreat within it. We have added the following clarification: "A minimum ice thickness of 15 m is maintained to preserve a thin ice shelf as it retreats."**

Eq. 3 Nitpicking, I know, but please don't use cursive in subscripts ($h\_af$).

**Thank you for your detailed suggestion. We have made the necessary modifications.**

L 123-124 "m is a positive exponent, often related to the creep exponent n of Glen's law (Glen, 1958) as m = 1/n. Here we use m = 3, following Hill et al. (2023)" This seems contradictory. Do you deviate from the "often" used relation of m=1/n (as typically n=3), or is there a typo somewhere and should I read m=1/3?

**We apologize for the confusion. The relation m = 1/n is incorrect in this context. Here, m directly corresponds to the creep exponent n in Glen's law (m=n=3). We use m instead of n for consistency with Joughin et al. (2019; https://doi.org/10.1029/2019GL082526). We have revised the text for clarity: "m is a positive exponent corresponding to the creep exponent in Glen's law (Glen 1958). Here, we use m = 3 following Joughin et al. (2019) and Hill et al. (2023)."**

L 145 Here too, if you use regular instead of cursive, the "JregEη2" term probably will look a lot cleaner.

**We agree with your suggestion and have made the modification.**

L 162-163 "...we initiate historical runs to smoothly transition the model past an initial adjustment phase in the forward transient simulations (Fig. 2). The historical runs span 20 years, from 1995 to 2015." If you really want to present your simulations as actual projections, this part will need more information. How exactly are your historical simulations forced in terms of atmosphere and ocean? How does your modelled trend in ice mass/thickness compare to observations? What year is the BedMachine dataset supposed to represent, and how does that affect the results, given that you used it to initialise your model in 1995?

**As mentioned earlier, we have removed the statement regarding quantitative projections. We found that adding extensive details here would distract from the main focus of the paper, so we have opted to retain the current level of detail.**

L 173-174 "In "sub-element melt 1" (SEM1), melt is applied to the entire area of partially floating elements, but its magnitude is reduced based on the fraction area of the floating ice in the element" Do I understand it correctly then that this is identical to the "partial melt parameterisation (PMP)" of Leguy et al. (2021) and Berends et al. (2023)?

**Yes, SEM1 is essentially identical to the PMP described by Leguy et al. (2021) and Berends et al. (2023). In our work, we have adopted the naming convention used by Seroussi and Morlighem (2018).**

L 175-177 "In the "sub-element melt 3" (SEM3), an increased number of 20 integration points are used during the finite element assembly procedure within any partially floating element" Does this mean that the entire stress balance is solved with a much higher resolution at the grounding line? If so, how does that affect the error in the velocity solution related to the discontinuous basal friction there?

**We have adopted sub-element parameterization 3 (SEP3) for resolving basal friction on partially floating elements, as discussed by Seroussi et al. (2014). SEP3 has demonstrated its superiority and has been widely adopted by most ice flow models. It allows the entire stress balance to be solved at a higher resolution around the grounding line. We believe that a detailed discussion of SEP3's impact is beyond the scope of this study.**

Eq. 7 Sorry for the nitpicking again, but I'd use a single-letter variable for SMB, since combined with the cursive font it now reads like "S times M times B".

**This notation is consistent with the naming convention of ISMIP6, as seen in Nowicki et al. (2020; https://doi.org/10.5194/tc-14-2331-2020). We have changed to using a regular font instead of cursive to make the function more intuitive and clear.**

Eq. 8 I think it would be really valuable to include a figure here comparing the spatial patterns of basal melt underneath one of the shelves with and without the water column scaling.

**We agree with your suggestion. We have added a comparison plot of the BMB distribution for the Cook Ice Shelf below Equation 8 to illustrate the differences with and without water column scaling.**

L 201-205 As with the historical simulations, if you wish to present your results as projections, you will need to provide more information here. Did you include the ISMIP6 "cavity-extrapolated ocean forcing"? What kind of temperatures does this produce in the very deep trenches in the Wilkes basin? Why did you use output from two different ocean models for SSP1 and SSP5?

**We have incorporated the ISMIP6 "cavity-extrapolated ocean forcing" in our study. It should be noted that the ocean temperatures in the deep trenches, based on this extrapolation, tend to be biased on the higher side. As previously mentioned, since**

we no longer consider our results as quantitative projections, we have chosen to maintain the text as it is.

**The ISMIP6-2300 project only provides UKESM thermal forcing data for SSP1, leaving us with no alternative. For the high emission scenario, we had two options from ISMIP6: UKESM and CESM2. We used both for the 500-year future simulations as initial tests and found that the UKESM-based results yielded a nonphysical high basal melt rate (over 150 m/a) when the grounding line retreats to the deep trenches. Comparatively, the CESM2-based results are closer to realistic melting rates, so we finally choice to use CESM2 for SSP5 instead of UKESM.**

L 206-211 Do I understand it correctly that you only performed the inversion with the Weertman law, and then converted the resulting friction coefficients to the Coulomb law to maintain the same basal friction?

**Yes, we performed the conversion from Weertman parameters to Coulomb parameters, as described in this paragraph.**

Fig. 10 No need to label every 15-year interval in the legend, maybe draw contours every 20 years but label only every 100? (apart from that, great figure!)

**We have experimented with wider intervals than 15 years, but we believe that the 15-year interval and labels are more effective. This interval allows readers to more easily identify the time point when the grounding line detaches from the deep trough and better reflects the geometric evolution of the glacier while maintaining the clarity of the plot. Therefore, we have decided to retain this figure as it is.**

Tables 3 & 4 That's a lot of numbers, consider replacing by a bar graph or something else to visualise.

**The data in Tables 3 and 4 are specific to Figure 13 (Convergence of total ice mass loss), so adding another bar chart would duplicate the information presented in Figure 13. However, your suggestion made us realize that these tables, with their extensive numerical data, may not be particularly useful to most readers and could impede reading fluency. Therefore, we have moved these tables to the appendix.**

Fig. 12 The graphs are suddenly much thicker, I like it! Please use these thick lines for the other figures too.

**Yes, we have modified the figures to use thicker lines for consistency.**

Fig. 13 Do you expect these lines to become straight when using a double-logarithmic scale?

**This is an excellent point. We have re-plotted the figures using a double-logarithmic scale. We believe this approach more effectively illustrates the convergence patterns and rates. We now present the results in this manner, showing an almost linear (i.e., constant convergence rate) relationship for each GLMP from 1000 to 500 to 250.**

L 325-326 "Due to the distinct mechanism of the model implementation, the GLMPs they used differ from the four explored in our study" Worth mentioning here that the resolutions used in these square-grid models are much coarser than what you used, so you would expect significant dependence on resolution there even if melt at the grounding line is resolved perfectly.

**Thank you for the suggestion. We have added this information accordingly.**

L 332-340 I think this is a crucial discussion. As far as I'm aware, all studies that have looked at GLMPs to date have implicitly assumed zero melt underneath grounded ice. If that assumption is wrong (as the studies you cite suggest), then obviously none of the simulations are ever going to get the "correct" answer, regardless of what GLMP they use. This is the "physics vs. model implementation" discussion I meant!

**Thank you for your affirmation. I completely agree with your opinion.**

All figures: please use a larger font size for the axis labels, legends, etc. Imagine you're in the back of the room at EGU!

**We have modified all figures to use thicker lines for improved readability.**

**Please note our responses are in bold.**

Summary:

The authors conduct a comprehensive study of ice dynamics sensitivity to grounding line melt parametrization, together with variations in mesh resolution, friction law and water column scaling, and emissions scenario. The study is focused on the Wilkes Subglacial Basin (WSB) region of East Antarctica. Experiments are run out to the year 2500 and grounding line dynamics as well as changes in total ice mass are evaluated.

Generally, melt parameterizations shows better convergence with resolution, except the NMP under the Coulomb law with water column scaling, where finer resolutions increase ice mass loss. This behavior is attributed to the NMP underestimating melt in partially grounded elements, which is inherent to the parameterization. SEM1 and SEM3, while providing more accurate average melt rates, do not necessarily improve convergence and often overestimate mass loss at coarse resolutions. Under high emission scenarios, differences in grounding line melt parameterizations performance are amplified, affecting ice mass loss predictions significantly. Overall, SEM and NMP outperform FMP, with each showing varying degrees of superiority depending on the scenario.

Overall, I think this study is well presented and is an interesting contribution with both location specific and general takeaways. It provides new and useful details on model dependence on parameterization and focuses on an important and understudied region of East Antarctica that needs more attention.

Major points:

1. Since this study is a sensitivity test, it would be really nice to see a comparison figure at the end showing all of the results. For example, this could display total ice mass over time (like existing plots) but show the results from melt parameterizations, emission scenarios, friction law, water column scaling, and model resolution. Currently, I find myself having to flip back and forth between all the result figures. A big comparison figure with all the results (or most important results) would help a lot with this.

**Thank you for your suggestion. While we appreciate the value of a comprehensive comparison figure, we believe that incorporating all the variables and results into a single graph would make it cluttered and potentially obscure the key points. The primary focus of this study is to evaluate the different GLMPs under various model scenarios, specifically through the convergence of total ice mass loss with model resolution. We have highlighted this critical result in the discussion section. Therefore, we believe an additional comparison figure is not essential and prefer to maintain the current format.**

2. As there is growing community interest in the WSB region, I would like to know what additional constraints the authors think would make the biggest difference for numerical simulations since this study shows that the melt parameterization affects both the timing of a tipping point and the overall magnitude of ice mass loss. This is sort of generally touched on but I think it should be expanded on in some more detail in the discussion.

**My co-authors and I respectfully differ in our perspective regarding this point. The focus of this paper is on exploring the technical implementation of basal melt in the model. The discussion of additional constraints on numerical simulations for the WSB extends beyond the scope of this study. As suggested by another reviewer, we have chosen not to regard our simulations as quantitative projections and have removed the relevant content. However, it is worth noting that we are currently conducting a separate study to couple SSA with subglacial hydrological systems. In that forthcoming paper, we will provide a more detailed and comprehensive scientific discussion of the current constraints on accurate quantitative projections of future ice mass loss.**

3. I would like to see something added about the choice of SSA over a higher order approximation of full stokes, since SSA has limitations in accurately representing grounding line dynamics. I understand that in order to run all of these experiments, SSA is probably the only computationally manageable option. But I would like to see the limitations addressed somewhere, i.e. is it possible that SSA could be inadequate for resolving processes that affect the results of this study?

**We completely agree with your concern that SSA might not be adequate for resolving grounding line processes. However, as mentioned earlier, we no longer treat this study as a quantitative projection of future ice mass change. Thus, the discussion of the limitations of SSA is beyond the scope of the current study. We have decided to retain the existing discussion.**

4. In the conclusion you say that the 1km grid isn't fine enough resolution for capturing the grounding line dynamics. Since it is so common to use even coarser resolutions than this for large scale ice sheet models, is there anything more the authors can add to this discussion? Would you say that high resolution should be prioritized above all else for modeling grounding line dynamics?

**In the conclusion section, we noted that "At a resolution considered high and commonly employed in ice sheet models (i.e., 1 km), numerical errors due to inadequate convergence can lead to an overestimation of mass loss by up to 40%**

**when compared to our finest mesh resolution of 250 m." The overestimation of ice loss at a 1 km resolution compared to 250 m does not directly imply that 1 km is insufficient for capturing grounding line dynamics. The reasons for this overestimation are complex, multifaceted, and not necessarily attributable solely to grounding line dynamics. Our intention is to objectively present our model results and to highlight to ice sheet modelers that results obtained at a resolution of 1 km, although recognized as fine, still differ from those obtained at a finer resolution of 250 m. We cannot definitively conclude that high resolution should be prioritized above all else for modelling grounding line dynamics.**

Minor points:

- I think it would be useful to have all the experiments introduced earlier in the paper. Right now only some of them are introduced early on and then others are introduced about half way through.

**Thank you for the suggestion. We have revised the Methods section to include an overview of the model components used in our sensitivity experiments at the beginning of the section.**

- L 84: could use a little more specifics rather than just the obvious results, discussion, conclusion.

**Yes, we have made the necessary modifications.**

- Fig 3: Add outline box in a) showing the region displayed in b)- e)

**Yes, this is an excellent suggestion. We have implemented it accordingly.**

- Fig 4: Units should be added

**The three variables presented in Figure 4 are dimensionless.**